# Investigating the Role of Intravoxel Incoherent Motion Diffusion-Weighted Imaging in Evaluating Multiple Sclerosis Lesions

**DOI:** 10.3390/diagnostics15101260

**Published:** 2025-05-15

**Authors:** Othman I. Alomair, Sami A. Alghamdi, Abdullah H. Abujamea, Ahmed Y. AlfIfi, Yazeed I. Alashban, Nyoman D. Kurniawan

**Affiliations:** 1Radiological Sciences Department, College of Applied Medical Sciences, King Saud University, P.O. Box 145111, Riyadh 4545, Saudi Arabia; salghamdi1@ksu.edu.sa (S.A.A.); ayf100@hotmail.com (A.Y.A.); yalashban@ksu.edu.sa (Y.I.A.); 2King Salman Centre for Disability Research, Riyadh 11614, Saudi Arabia; 3Department of Radiology and Medical Imaging, King Saud University Medical City & College of Medicine, King Saud University, Riyadh 7805, Saudi Arabia; abujamea@ksu.edu.sa; 4Radiology Department, Maternity and Children’s Hospital in Dammam Eastern Health Cluster, Dammam, Saudi Arabia; 5Centre for Advanced Imaging, Australian Institute for Bioengineering and Nanotechnology, The University of Queensland, Brisbane, QLD 4072, Australia; nyoman.kurniawan@cai.uq.edu.au

**Keywords:** multiple sclerosis (MS), MS lesions, intravoxel incoherent motion (IVIM), apparent diffusion coefficient (ADC), pure molecular diffusion (*D*), pseudo diffusion (*D**), perfusion fraction (*f*)

## Abstract

**Background**: Multiple sclerosis (MS) is a chronic and heterogeneous disease characterized by demyelination and axonal loss and damage. Magnetic resonance imaging (MRI) has been employed to distinguish these changes in various types of MS lesions. **Objectives:** We aimed to evaluate intravoxel incoherent motion (IVIM) diffusion and perfusion MRI metrics across different brain regions in healthy individuals and various types of MS lesions, including enhanced, non-enhanced, and black hole lesions. **Methods:** A prospective study included 237 patients with MS (65 males and 172 females) and 29 healthy control participants (25 males and 4 females). The field strength was 1.5 Tesla. The imaging sequences included three-dimensional (3D) T_1_, 3D fluid-attenuated inversion recovery, two-dimensional (2D) T_1_, T_2_-weighted imaging, and 2D diffusion-weighted imaging (DWI) sequences. IVIM-derived parameters—apparent diffusion coefficient (ADC), pure molecular diffusion (*D*), pseudo-diffusion (*D**), and perfusion fraction (*f*)—were quantified for commonly observed lesion types (2506 lesions from 224 patients with MS, excluding 13 patients due to MRI artifacts or not meeting the diagnostic criteria for RR-MS) and for corresponding brain regions in 29 healthy control participants. A one-way analysis of variance, followed by post-hoc analysis (Tukey’s test), was performed to compare mean values between the healthy and MS groups. Receiver operating characteristic curve analyses, including area under the curve, sensitivity, and specificity, were conducted to determine the cutoff values of IVIM parameters for distinguishing between the groups. A *p*-value of ≤0.05 and 95% confidence intervals were used to report statistical significance and precision, respectively. **Results:** All IVIM parametric maps in this study discriminated among most MS lesion types. ADC, *D*, and *D** values for MS black hole lesions were significantly higher (*p* < 0.0001) than those for other MS lesions and healthy controls. ADC, *D*, and *D** maps demonstrated high sensitivity and specificity, whereas *f* maps exhibited low sensitivity but high specificity. **Conclusions:** IVIM parameters provide valuable diagnostic and clinical insights by demonstrating high sensitivity and specificity in evaluating different categories of MS lesions.

## 1. Introduction

Multiple sclerosis (MS) is a chronic inflammatory demyelinating disease of the central nervous system (CNS), which is characterized by a wide range of clinical manifestations and pathological changes. MS is one of the most common neurological disorders affecting young adults, often leading to significant disability over time [1,2]. The disease pathology includes varying degrees of inflammation, demyelination, axonal loss, and gliosis, contributing to the heterogeneity observed in clinical presentations and disease progression [3].

Most patients with MS are diagnosed with relapsing-remitting multiple sclerosis (RR-MS), which primarily affects young adults, with an average age of symptom onset of 30 years and a higher incidence in women. RR-MS is characterized by episodes of neurological impairment or relapses, interspersed with periods of symptom-free remission [4]. While the exact mechanisms underlying MS development are not fully understood, increasing evidence suggests that a vascular component may play a role in disease progression [5,6,7]. Evidence of vascular involvement includes the formation of MS lesions primarily around central veins, metabolic dysfunction resulting from hypoperfusion, and the identification of microvascular occlusions, indicative of ischemic conditions [7,8].

In magnetic resonance imaging (MRI), MS lesions can appear as black hole, enhanced lesions, or non-enhanced lesions. Black hole lesions, which are hypointense on T_1_-weighted images (T_1_WI), represent areas of severe axonal loss and tissue destruction. These lesions are associated with long-term disability and reflect the chronic, irreversible damage observed in advanced stages of the disease [9]. Enhanced MS lesions appear hyperintense on T_1_WI following the administration of gadolinium contrast agent, indicating active inflammation and blood–brain barrier disruption [10,11,12]. These lesions are typically associated with ongoing disease activity and relapses. Non-enhanced MS lesions are those that do not exhibit gadolinium enhancement and usually represent older, more stable areas of damage where the acute inflammatory process has subsided [3]. MS research requires imaging techniques that are both sensitive and reliable to examine specific pathological changes in the white matter as the disease progresses [11].

MRI is the predominant tool for diagnosing MS. Conventional MRI techniques, such as T_1_WI, are used to detect chronic neurodegeneration [13], while positive gadolinium-enhanced contrast enhancement highlights immune cell migration across the blood–brain barrier and active inflammation in lesions [14,15]. T_2_-weighted imaging (T_2_WI) is frequently utilized to identify and measure the number and volume of clinically silent lesions [16]. However, conventional MRI techniques lack the sensitivity and specificity required to fully assess the extent and severity of the diverse lesions observed in MS. Furthermore, a poor correlation exists between lesion quantification using MRI and functional disabilities assessed using clinical measures [17,18,19].

Gadolinium-based contrast agents are widely employed in MRI to highlight active MS lesions by identifying areas of blood–brain barrier disruption [11,12]. These agents are vital for diagnosing and monitoring disease progression and evaluating treatment efficacy in patients with MS. However, the use of gadolinium-based contrast agents has recently come under scrutiny owing to studies reporting gadolinium accumulation in the brain. This issue is particularly significant for the MS population, as patients frequently undergo repeated MRI involving gadolinium-based agents [13,20].

Among advanced MRI techniques, intravoxel incoherent motion (IVIM) has emerged as a promising tool for providing detailed insights into the microstructural alterations associated with different stages of MS [21,22]. IVIM MRI differentiates between pure molecular diffusion (*D*), pseudo-diffusion due to perfusion in capillaries (*D**), and perfusion fraction (*f*), thereby offering a comprehensive assessment of both diffusion and perfusion within the tissue [23,24]. This technique enables the simultaneous quantification of microcirculatory perfusion and tissue diffusion, providing unique insights into the pathophysiological processes of neurological diseases, including the formation of active, non-active, and black hole lesions [25].

In liver diseases, IVIM has proven valuable for non-invasive evaluation of hepatic inflammation. The IVIM-*f* parameter has been shown to accurately distinguish varying levels of inflammation, demonstrating its potential as a biomarker for the severity and extent of hepatic inflammation [26]. In oncology, IVIM parameters—*D*, *D**, and *f*—have been particularly useful in differentiating between benign and malignant tumors [27]. Thus, the integration of IVIM MRI into the study of MS offers a promising approach to enhance our understanding of this complex disease by providing detailed microstructural insights into the heterogeneous nature of MS lesions [21]. This advanced understanding could facilitate the identification of novel biomarkers for early diagnosis, improved monitoring of disease progression, and the development of targeted therapeutic strategies.

Despite its potential, the application of IVIM in MS research remains limited and is still in its nascent stages compared to its established use in fields such as oncology and hepatology. The scarcity of studies specifically focusing on IVIM in MS highlights a significant research gap. Much remains to be explored to fully understand its benefits for diagnosing and characterizing MS, particularly in detecting and evaluating perfusion changes associated with the disease.

This study aimed to assess and compare diffusion and perfusion MRI metrics, which do not require contrast agents, across different brain regions in healthy individuals and various types of MS lesions, including enhanced, non-enhanced, and black hole lesions. By examining these differences, the study seeks to identify potential diagnostic biomarkers capable of distinguishing normal brain tissue from various MS lesion types.

## 2. Materials and Methods

### 2.1. Sample Criteria

A total of 237 patients with MS (65 males and 172 females) and 29 healthy control participants (25 males and 4 females) were recruited for MRI scanning at King Khalid University Hospital, Riyadh, Saudi Arabia. Ethical approval was obtained from the Local Ethics Committee at King Khalid University Hospital, Riyadh, Saudi Arabia (No. E-23-7517), and written informed consent was obtained from all participants. This prospective study was conducted from January 2019 to December 2020.

The sample size in our experiment was assumed to be the number of regions of interest (ROIs). This is a reasonable assumption because it permits comparing between similarly located ROIs in healthy subjects and MS, as the lesions were found to aggregate in various brain structures [28]. Such an approach was found useful to improve the assessment accuracy and better understanding of microstructural changes in both healthy tissues and all MS lesions, such as axonal loss [29,30].

Our experiment included 29 controls with the IVIM parameters measured using a total of 290 ROIs in various brain structures, including the corpus callosum (CC), white matter (WM), and deep WM. These ROIs were chosen to compare types of MS lesions in specific brain areas, which are highly myelinated and common sites for MS lesions [31,32]. The ROI selection in the CC allowed evaluation of brain connectivity between interhemispheric structure [33] and provided a reference for evaluating either demyelination or axonal injury of the nervous tissues [29,30]. The composition of ROIs in controls are as follows: 58 ROIs in the CC, comprised of two ROIs at the genu and splenium; 116 ROIs in the WM, comprised of two ROIs in the right and two ROIs in the left frontal WM areas; and 116 ROIs in the deep WM, comprised of four ROIs located in the right and the left thalamus and in the right and the left caudate nuclei.

Among the 237 patients, 224 were diagnosed with MS based on the McDonald criteria [11,34], which require evidence of lesion dissemination in both space and time. These patients were classified as having RR-MS with a mean age of 36.25 ± 9.7 years and a mean disease duration of 8.9 ± 4.7 years. Thirteen patients with MS were excluded owing to either motion artifacts compromising imaging quality or not meeting the diagnostic criteria for RR-MS. The control group consisted of healthy individuals without neurological disorders, with a mean age of 28 ± 4.7 years. Table 1 summarizes demographic and clinical data for controls and RR-MS patients.

### 2.2. Lesion Analysis

Three MS lesion patterns were analyzed, including enhanced, non-enhanced, and black holes, as shown in Figure 1. MS enhanced (MS_E) lesions were identified as hyperintense on post-contrast T_1_WI and hyperintense on T_2_WI. These lesions were observed in 42 patients with MS, with a total of 114 lesions. MS non-enhanced (MS_NE) lesions appeared isointense on both pre- and post-contrast T_1_WI and hyperintense on T_2_WI, with 1613 lesions identified across all patients with MS. MS black hole (MS_BH) lesions were characterized by hypointensity on both pre- and post-contrast T_1_WI and hyperintensity on T_2_WI, totaling 779 lesions. Of these, 119 black hole lesions were found in patients with enhanced lesions, while 660 were identified in patients with non-enhanced lesions.

### 2.3. MRI Imaging Protocols

MRI brain examinations were performed using a 1.5-Tesla GE MRI scanner with a magnetic gradient strength of 45 mT/m (GE Healthcare, Waukesha, WI, USA) and Syngovia software version LX 27 (Siemens Healthineers, Erlangen, Germany), multichannel phased-array head coil equipped with eight channels. All patients with MS underwent standard imaging protocols for MS, which included the following imaging sequences: three-dimensional (3D) T_1_, 3D fluid-attenuated inversion recovery (FLAIR), two-dimensional (2D) T_2_WI, and 2D diffusion-weighted imaging (DWI), in addition to 2D and 3D post-contrast T_1_WI imaging. Three-dimensional pre- and post-contrast T_1_WI images were acquired using the Fast Spoiled Gradient “FSPGR” sequence with the following parameters: repetition time (TR) = 9 ms, echo time (TE) = 4 ms, and an acquired resolution of 1.8 × 0.5 × 0.5 mm. Three-dimensional FLAIR images were obtained using the Fast Spin Echo “FSE” sequence with the following parameters: TR = 6900 ms, TE = 94 ms, and an acquired resolution of 0.9 × 0.5 × 0.5 mm. Two-dimensional axial T_2_WI images were acquired using the Blade FSE sequence with the following parameters: TR = 7807 ms and TE = 123 ms. The 2D post-contrast T_1_WI images were acquired using the Spin Echo “SE” sequence with the following parameters: TR = 517 ms and TE = 9 ms. Both 2D T_2_WI and 2D post-contrast T_1_WI were acquired with the same resolution of 0.4 × 0.4 × 4.5 mm. Additionally, DWI images were acquired using the single-shot echo-planar imaging technique for patients with MS as well as control participants. The acquisition settings included TR = 4508 ms, TE = 77.5 ms, and an acquired resolution of 0.8 × 0.8 × 5 mm. Seven b-values (0, 30, 50, 70, 100, 200, 500, and 1000 s/mm^2^) were used in three orthogonal directions to generate a mono-exponential apparent diffusion coefficient (ADC) map and to produce *D*, *D**, and *f* maps using the segmented IVIM mode. Two-dimensional conventional T_2_ WI, post-contrast T_1_WI, and DWI sequences were acquired with the same slice location. Figure 2 presents a flowchart of the imaging protocol and processing steps.

### 2.4. MRI Processing

All images were processed by two consultant neuroradiologists. ADC and IVIM maps were generated using the OsiriX-plugin IB-Diffusion™ software, version 21.12 (Imaging Biometrics, Elm Grove, WI, USA) [35]. The segmented IVIM model was employed to quantify *D, D**, and *f* values. In this experiment, the *b_inflection_* point, defined as the end of the perfusion influence range, was set at 200 s/mm^2^. Using the segmented IVIM model, *D** was calculated using *b*-values below the *b_inflection_* point, while *D* was calculated using *b*-values above the *b_inflection_* point. The *f* values were determined using the following equation:
(1)
$$f=\frac{S_{0}-S_{extrapolated(0)}}{S_{0}}$$


Here, *S*_0_ represents the measured signals without diffusion gradients applied, and *S*_extrapolated(0)_ represents the measured signals with the minimum *b*-value *b_inflection_* set at the binflection point. ADC, *D*, and *D** values were expressed in units of 10^−3^ mm^2^/s.

DWI datasets were acquired using 2D EPI at 0.8 × 0.8 × 5 mm resolution, and 2D T_1_- and T_2_-weighted images were acquired at 0.4 × 0.4 × 4.5 mm resolution. DWI had a lower image resolution than the T_1_WI and T_2_WI to minimize EPI distortion artefacts. DWI motion correction was applied by the scanner during the scan. For each patient, the ADC and IVIM maps derived from the DWI and b_0_ images were already co-registered.

Image registration was performed on all patient data to ensure all MR images were correctly aligned. This is important because some patients were repositioned after contrast media administration (Figure 3), and image registration would increase the accuracy of defining MS lesions and facilitated comparison between the T_1_WI and T_2_WI, as well as the IVIM parametric maps.

Post-contrast 2D T_1_WI was used as a reference to register the image dataset for each patient. First, the T_2_WI was registered to T_1_WI using a rigid body followed by an affine registration. Second, the T_2_ WI was then registered to DWI’s b_0_ image as they had similar contrast. This transformation was then applied to the T_1_WI, so that all images were all in the same space. A correlation ratio cost function was applied for the multi-contrast MRI registration [36,37,38], and a tri-linear image interpolation was used to smooth the registered images for analysis. This step was critical in identifying three types of MS lesions: enhanced, non-enhanced, and black hole.

A similar registration approach was utilized for both control subjects and MS patients. However, for control subjects, we used b_0_ as a reference image because 2D T_1_WI and T_2_WI were not acquired for healthy participants. Figure 4 demonstrates the registration of b_0_ to all IVIM parametric maps.

Linear registration FLIRT FSL helped to limit rotation, scaling, shearing, and translation, which might occur due to patient movement or image distortion. This registration method was appropriate because all images were registered within the same subject. This in turn improved the experiment’s workflow and ROI selection in the same 2D structural images and 2D IVIM parametric maps for both control and MS patients. This consequently allowed us to quantify ADC, *D*, *D**, and *f* values using the IVIM parametric maps in both control and MS patients by reading out all ROIs using ITK-SNAP [39], as shown in Figure 5 and Figure 6.

Regions of interest (ROIs) for control participants were delineated using b_0_ images with the ITK-SNAP program [39]. Four ROIs were outlined for control participants, including the genu of the corpus callosum (GCC) and the splenium of the corpus callosum (SCC), both marked in red; the white matter, marked in blue; and the deep white matter, marked in yellow, as shown in Figure 5. Measurements from the GCC and SCC were averaged and collectively referred to as the corpus callosum (CC). Similarly, measurements from the right and left frontal white matter areas, as well as the right and left frontoparietal white matter at the convexity, were averaged and collectively referred to as the WM. Measurements from the right and left thalamus and the right and left caudate nuclei were averaged and collectively referred to as the deep white matter. The size of CC ROIs was maintained at 600 ± 50 mm^2^, while the sizes of the white matter and deep white matter ROIs were maintained at 284 ± 5 mm^2^.

For patients with MS, lesion ROIs were delineated using 2D post-contrast T_1_WI with the ITK-SNAP program (Figure 6). Typically, an ROI was drawn in the central area of an MS lesion on the T_1_WI and subsequently registered onto the 2D T_2_WI, ADC, and IVIM parametric maps. ROIs were circular for most MS lesions, with sizes varying based on the lesion shape. All delineated ROIs were visually confirmed by two radiologists (MSA and HMA).

### 2.5. Statistical Analysis

Data were analyzed using IBM SPSS Statistics for Windows, version 26.0 (IBM Corp., Armonk, NY, USA), and MedCalc Statistical Software, version 19.2.6 (MedCalc, Ostend, Belgium). Descriptive statistics, including mean, standard deviation, minimum, and maximum values, were used to summarize the outcome variables (ADC, *D*, *D**, and *f*). Results are presented as group means ± standard deviation.

A one-way analysis of variance (ANOVA), followed by Tukey’s post-hoc test, was conducted to compare mean values between MS lesion types (MS_E, MS_NE, and MS_BH) and healthy control regions (CC, WM, and deep WM). Receiver operating characteristic (ROC) curve analysis was performed to determine cutoff values for the outcome variables, utilizing area under the curve (AUC), sensitivity, and specificity measurements. A *p*-value of ≤0.05 and 95% confidence intervals were used to report the statistical significance and precision of the results, respectively.

## 3. Results

### 3.1. Descriptive Statistics

The descriptive statistics for the outcome variables are summarized in Table 2 and Figure 7. For the ADC values, the mean measurements in control regions were as follows: CC (0.75 ± 0.04 mm^2^/s), white matter (0.75 ± 0.04 mm^2^/s), and deep white matter (0.73 ± 0.03 mm^2^/s). Among MS lesions, enhanced lesions exhibited an ADC of 0.97 ± 0.16 mm^2^/s, non-enhanced lesions had 1.06 ± 0.16 mm^2^/s, and black holes demonstrated the highest value of 1.29 ± 0.21 mm^2^/s (*p* < 0.0001), significantly higher than controls and other MS lesion types.

For the *D* values, the mean measurements in control regions were as follows: CC (0.72 ± 0.04 mm^2^/s), white matter (0.71 ± 0.04 mm^2^/s), and deep white matter (0.71 ± 0.03 mm^2^/s). For MS lesions, enhanced lesions had a mean of 0.92 ± 0.16 mm^2^/s, non-enhanced lesions measured 1.01 ± 0.16 mm^2^/s, and black holes lesions displayed the highest value at 1.23 ± 0.22 mm^2^/s (*p* < 0.0001) compared to controls and other MS lesion types.

For *D** values, the mean values in control regions were as follows: CC (0.87 ± 0.09 mm^2^/s), white matter (0.89 ± 0.08 mm^2^/s), and deep white matter (0.86 ± 0.11 mm^2^/s). In MS lesions, enhanced lesions had a mean *D** of 1.17 ± 0.25 mm^2^/s, non-enhanced lesions measured 1.22 ± 0.19 mm^2^/s, and black holes lesions demonstrated the highest value at 1.47 ± 0.22 mm^2^/s (*p* < 0.0001), significantly exceeding those of controls and MS enhanced lesions.

For *f*, the control region means were as follows: CC (0.04 ± 0.01), white matter (0.05 ± 0.01), and deep white matter (0.04 ± 0.02). Among MS lesions, enhanced lesions had a mean *f* value of 0.07 ± 0.04, non-enhanced lesions measured 0.06 ± 0.02, and black holes displayed a value of 0.06 ± 0.03.

### 3.2. One-Way ANOVA

The one-way ANOVA analysis (Table 3) revealed statistically significant differences in the mean values of ADC, *D*, *D**, and *f* among the control and MS lesion groups. The F-values and *p*-values for each parameter are as follows: ADC: F = 482.52, *p* < 0.0001; *D*: F = 433.60, *p* < 0.0001; *D**: F = 431.40, *p* < 0.0001; and *f*: F = 25.77, *p* < 0.0001.

### 3.3. Multiple Comparisons of MRI Parameters

#### 3.3.1. Control Regions vs. MS Lesions

Multiple comparisons of ADC, *D*, *D**, and *f* between control regions (CC, white matter, and deep white matter) and MS lesions (enhanced, non-enhanced, and black holes) revealed significant differences. For ADC, *D*, and *D**, the mean values in control regions were significantly lower than those observed in all types of MS lesions (*p* < 0.0001 for all comparisons). Similarly, for *f*, control regions showed significantly lower mean values compared to MS lesions, with significance levels of *p* < 0.0001 for enhanced lesions, *p* = 0.002 for non-enhanced lesions, and *p* < 0.0001 for black holes. Detailed comparisons are presented in Table 4. The Tukey’s test *p*-values are shown in Table 4.

#### 3.3.2. Between MS Lesions

Among the different types of MS lesions, multiple comparison analyses identified significant differences in ADC, *D*, *D**, and *f* parameters. Enhanced lesions exhibited significantly lower ADC values compared to non-enhanced lesions and black holes (*p* < 0.0001 for both), while non-enhanced lesions had lower ADC values than black holes (*p* < 0.0001). For *D*, enhanced lesions showed significantly lower values than both non-enhanced lesions and black holes (*p* < 0.0001 for both), and non-enhanced lesions also demonstrated lower values than black holes (*p* < 0.0001). For *D**, enhanced lesions displayed significantly lower values compared to black holes (*p* < 0.0001), but their values were not significantly different from non-enhanced lesions (*p* = 0.264). Non-enhanced lesions had lower *D** values than black holes (*p* < 0.0001). For *f*, enhanced lesions had higher values than non-enhanced lesions (*p* < 0.0001), but their values were not significantly different from black holes (*p* = 0.202). Non-enhanced lesions had lower *f* values than black holes (*p* < 0.0001). Detailed comparisons are presented in Table 5. The Tukey’s test *p*-values are shown in Table 5.

### 3.4. ROC Curve Analysis

The ROC curve analysis identified cutoff values for ADC, *D*, *D**, and *f*, demonstrating high sensitivity and specificity in differentiating MS lesions from control regions, as summarized in Table 6. The following values were determined to distinguish between BH vs. controls, E vs. controls, and non-enhanced vs. controls. For ADC, the cutoff values were >0.84, >0.83, and >0.83, with sensitivity values of 98.0%, 89.5%, and 94.5% and specificity values of 99.7%, 98.6%, and 98.3%, respectively. For *D*, the cutoff values were >0.81, >0.78, and >0.79, with sensitivity values of 97.7%, 89.5%, and 93.6% and specificity values of 99.6%, 96.6%, and 98.3%, respectively. For *D**, the cutoff values were >1.08, >0.98, and >1.0, with sensitivity values of 95.3%, 86.0%, and 88.5% and specificity values of 99.0%, 89.7%, and 92.8%, respectively. For *f*, the cutoff values were >0.05, >0.05, and >0.05 with sensitivity values of 56.7%, 61.4%, and 49.8% and specificity values of 80.0%, 80.7%, and 80.0%, respectively. These findings underscore the potential utility of diffusion and perfusion MRI metrics in distinguishing MS lesions from healthy brain tissues, offering valuable diagnostic and clinical insights.

## 4. Discussion

Despite the limited number of studies available, IVIM MRI has demonstrated promise in advancing our understanding of MS pathology [21]. This study evaluated IVIM parametric maps across three categories of MS lesions and compared them with different brain regions in control participants, an area not previously reported in the literature. The results reveal significant differences in diffusion and perfusion metrics between control regions and various MS lesion types, highlighting substantial variance between the groups. These findings reinforce the potential utility of these parameters as diagnostic biomarkers for MS [22].

### 4.1. Comparison Between Controls and MS Lesions

The ADC values were significantly higher in MS lesions compared to control regions, consistent with previous studies reporting elevated ADC values in both acute and chronic MS lesions due to increased water diffusion associated with tissue damage and loss of cellular integrity [40,41]. For example, the mean ADC for black hole MS lesions is significantly higher than that of normal white matter, likely reflecting increased extracellular space and reduced cellular density resulting from severe tissue destruction [40].

IVIM-*D*, which reflects the diffusion motion of pure water molecules within the tissue of interest [42], also showed significant differences between control regions and MS lesions. The IVIM-*D* values for black hole MS lesions were markedly higher than those for controls and other MS lesions, suggesting pronounced diffusion changes and severe microstructural damage [43,44]. Enhanced MS lesions and non-enhanced MS lesions similarly exhibited higher IVIM-*D* values than control regions, indicating altered tissue integrity in these lesion types.

IVIM-*D**, the pseudo-diffusion coefficient, represents dephasing caused by perfusion in semi-randomly organized capillaries [42]. In healthy control regions such as the CC, white matter, and deep white matter, IVIM-*D** values were relatively low. By contrast, black hole MS lesions displayed significantly higher IVIM-*D** values, indicative of severe microstructural alterations characterized by extensive demyelination and axonal loss. Enhanced MS lesions and non-enhanced MS lesions also exhibited elevated IVIM-*D** values compared to controls, suggesting significant microstructural damage in these lesion types [43,45,46].

The IVIM-*f* values, which represent the perfusion fraction, were higher in MS lesions compared to control regions. Among the lesion types, enhanced MS lesions exhibited the highest IVIM-*f* values, suggesting increased microvascular density or perfusion fraction due to inflammatory processes and angiogenesis associated with MS pathology [7,47]. Black hole and non-enhanced MS lesions also demonstrated higher IVIM-*f* values compared to control regions, reflecting variations in perfusion characteristics among the lesion types [7].

This study revealed significant differences in diffusion and perfusion MRI metrics across the MS lesion types, highlighting substantial variations in the underlying pathology of MS. In contrast, a study by Johnson et al. [21], which employed IVIM MRI to assess changes in the spinal cord of patients with MS compared to healthy individuals, reported no significant differences in IVIM perfusion fraction or pseudo-diffusion measurements. However, their use of a small sample size and thick MRI slices (resulting in a large partial volume effect) may have reduced the sensitivity of IVIM MRI in detecting pathological changes.

The ROC curve analysis for ADC, *D*, *D**, and *f* metrics demonstrated their effectiveness in distinguishing MS lesion types from healthy controls. Metrics such as ADC, *D*, and *D** showed AUC values above 0.90, indicating excellent diagnostic performance with high sensitivity and specificity. These metrics consistently differentiated MS lesions from controls, underscoring their robustness as diagnostic tools. Although the *f* values had a lower AUC, they still provided high specificity, making them valuable for confirming lesion presence. Collectively, these findings support the use of ADC, *D*, and *D** metrics in clinical practice to accurately identify MS lesions.

### 4.2. Comparisons Between MS Lesion Types

Multiple comparisons between different types of MS lesions revealed significant differences in diffusion and perfusion MRI metrics. For ADC values, enhanced MS lesions exhibited significantly lower values compared to non-enhanced MS lesions, suggesting that non-enhanced lesions may experience greater water diffusion, likely due to more extensive tissue damage. ADC values were markedly higher in black hole MS lesions compared to both enhanced and non-enhanced MS lesions, indicating that black hole lesions are associated with the most severe tissue destruction and increased extracellular space [40,46].

The IVIM-*D* values showed a distinct pattern. Non-enhanced MS lesions had higher IVIM-*D* values than enhanced MS lesions, while black hole MS lesions exhibited the highest IVIM-*D* values, significantly exceeding those of both enhanced and non-enhanced MS lesions. These elevated IVIM-*D* values in black hole lesions reflect pronounced microvascular abnormalities and diffusion changes indicative of severe microstructural damage.

For IVIM-*D** values, no significant difference was observed between enhanced and non-enhanced MS lesions. However, black hole MS lesions exhibited significantly higher IVIM-*D** values than both enhanced and non-enhanced MS lesions, indicating severe microstructural alterations characterized by extensive demyelination and axonal loss.

The IVIM-*f* values were higher in enhanced MS lesions compared to non-enhanced MS lesions, suggesting increased microcirculatory perfusion in enhanced lesions. The IVIM-*f* values in enhanced MS lesions were slightly lower than those in black hole MS lesions, but this difference was not statistically significant. Non-enhanced MS lesions exhibited higher IVIM-*f* values compared to black hole MS lesions, suggesting differences in perfusion characteristics among lesion types [46].

In summary, our study demonstrated that ADC and IVIM parametric maps effectively differentiate healthy brain tissues from MS lesions and distinguish among various types of MS lesions (enhanced, non-enhanced, and black hole). This sensitivity suggests that these parameters may serve as potential diagnostic biomarkers, aiding in the identification and characterization of MS lesions, thereby enhancing diagnostic accuracy and clinical management.

Our findings indicate that IVIM MRI, which does not require gadolinium-based contrast agents, could offer a safer alternative for imaging MS lesions. This technique addresses concerns regarding gadolinium deposition in brain tissues while providing valuable insights into the microstructural and microvascular environment of MS lesions [11,20].

Despite these promising findings, our study has some limitations. First, the study was conducted at a single point in time, limiting the ability to assess the progression of MS lesions or the dynamic changes in DWI or IVIM parameters over time. Longitudinal studies are needed to evaluate these temporal changes. Second, the study was conducted at a single center, which may restrict the generalizability of the results to other settings. Multi-center studies are essential to validate these findings across diverse populations and clinical environments. Future research should aim to include a larger sample size, longitudinal follow-up, and multi-center participants to further validate these results. Additionally, such studies could explore the long-term clinical impact of these findings, providing a more comprehensive understanding of the role of diffusion and perfusion MRI metrics in MS diagnosis and management.

Third, 3D acquisitions were not used to delineate ROI MS lesions in our experiment because no 3D images were obtained for control subjects due to time constraints when using clinical scanners. This is one of the study’s limitations, as 3D pre- and post-contrast T1 WI and 3D FLAIR would help detect more MS lesions, particularly in the cortical and subcortical areas [48,49,50]. This could be used in future studies to classify MS lesions and quantify IVIM values in all brain regions. Last, the smaller control group sample size was primarily a practical limitation due to the challenges in recruiting healthy volunteers for advanced MRI protocols. Nevertheless, the preliminary findings from this study provide a valuable foundation for future work, where we aim to increase the control sample size to validate and expand upon these results.

Although our cohort analysis demonstrates robust discrimination of MS types from the control group (Table 4), individual lesion-level accuracy may be constrained by several factors. First, MS plaques progress pathologically from acute inflammatory to chronic demyelinated or remyelinated stages, resulting in diffusion-perfusion patterns that may overlap with those of control tissue [51]. Second, partial volume effects in small or irregularly shaped plaques can distort parameter contrasts [52]. Finally, the biexponential IVIM model is sensitive to noise at high b-values and is limited by a low-b sampling scheme, which can impair the precision of *D* and *f* estimates in lesions with subtle microvascular changes [21].

In our study, MS patients had been carefully selected based on the McDonald criteria [11,34] and without other detectable neurological disorders. There may be, however, some overlaps of IVIM signatures in MS lesions compared to brain tumor lesions or other lesions mimicking MS [53]. Future studies could investigate the specificity of IVIM for differentiating MS lesions from other types of brain lesions [54,55].

## 5. Conclusions

Our study showed that the ADC and IVIM *D* parameters were sensitive and specific markers to distinguish between normal white matter brain tissues and different types of MS lesions. The role of perfusion IVIM *D** and IVIM *f*, however, appeared to be more limited. Further studies are needed to determine if ADC and IVIM can reliably be used, instead of contrast agents, to distinguish normal brain tissue from different MS lesion types.

## Figures and Tables

**Figure 1 diagnostics-15-01260-f001:**
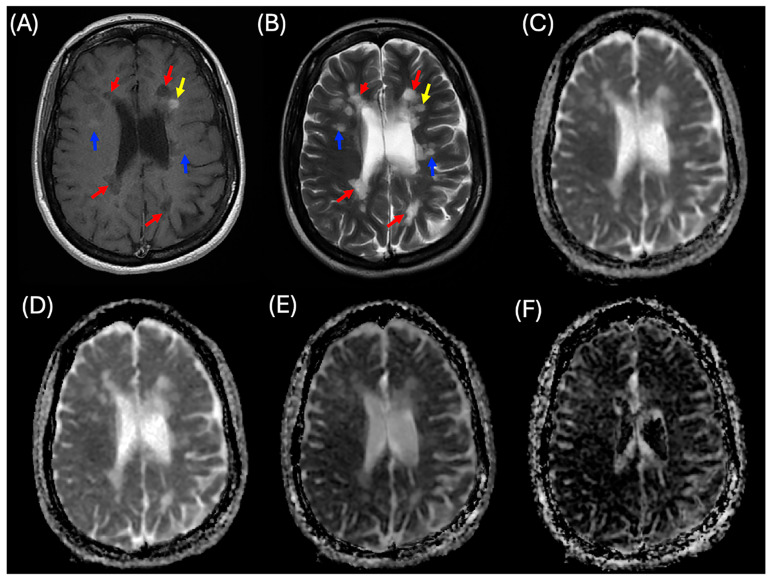
A case of a patient with relapsing-remitting multiple sclerosis (MS) illustrating three categories of MS lesions. (**A**) Two-dimensional (2D) post-contrast T_1_-weighted imaging (T_1_WI), (**B**) 2D T_2_-weighted imaging (T_2_WI), (**C**) apparent diffusion coefficient (ADC) map, (**D**) diffusion coefficient (*D*) map, (**E**) pseudo-diffusion coefficient (*D**) map, and (**F**) perfusion fraction (*f*) map. Yellow arrows indicate enhanced MS lesions, blue arrows indicate non-enhanced MS lesions, and red arrows indicate black hole lesions.

**Figure 2 diagnostics-15-01260-f002:**
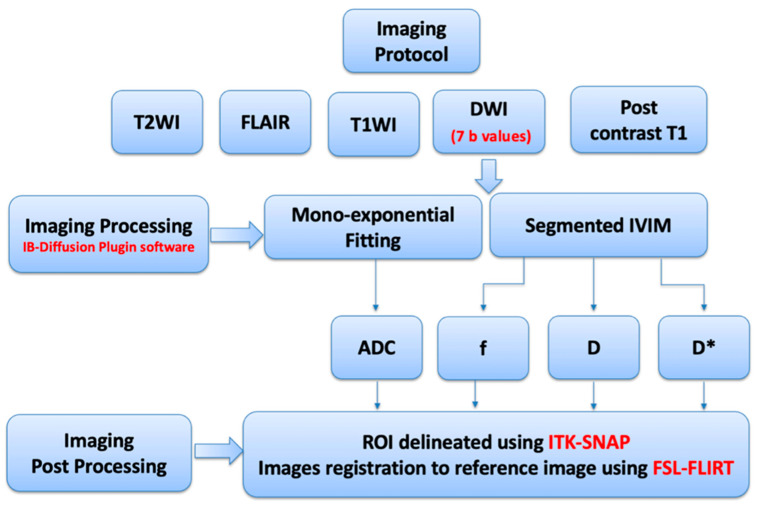
Flowchart illustrating the imaging protocol and processing steps.

**Figure 3 diagnostics-15-01260-f003:**
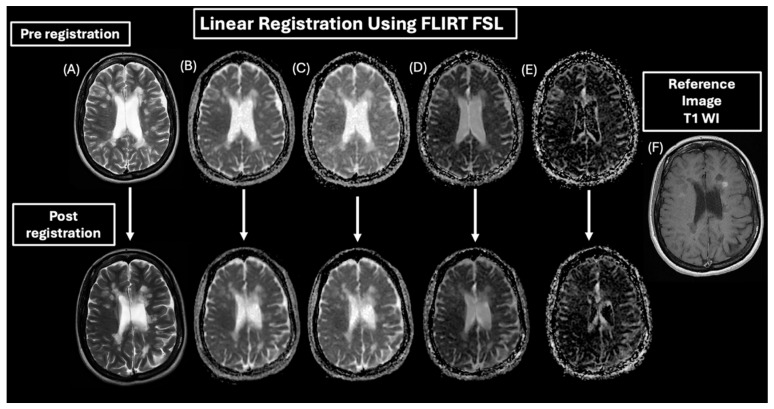
Pre- and post-linear registration of the IVIM parametric maps of a MS patient. (**A**) T2 WI, (**B**) ADC map, (**C**) diffusion coefficient (*D*) map, (**D**) pseudo-diffusion coefficient (*D**) map, and (**E**) perfusion fraction (*f*) map to the T1WI as a reference image (**F**).

**Figure 4 diagnostics-15-01260-f004:**
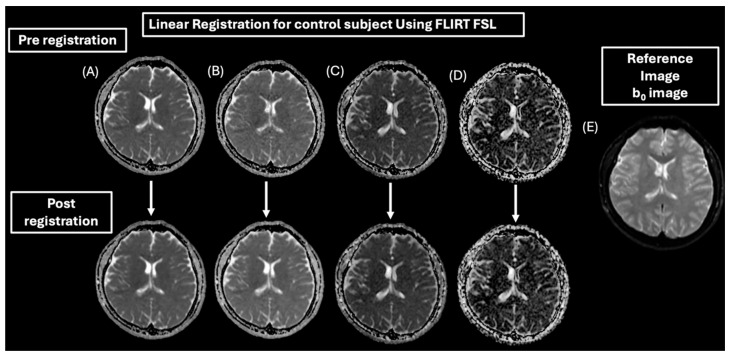
Pre- and post-linear registration of the IVIM parametric maps of a control subject. (**A**) ADC map, (**B**) diffusion coefficient (*D*) map, (**C**) pseudo-diffusion coefficient (*D**) map, and (**D**) perfusion fraction (*f*) map to the b0 as a reference image (**E**).

**Figure 5 diagnostics-15-01260-f005:**
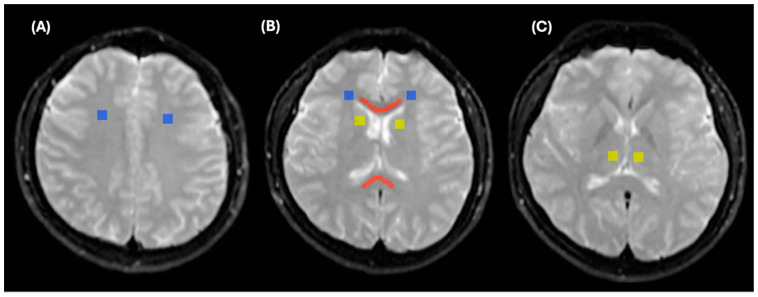
Regions of interest (ROIs) for control participants, delineated on b_0_ images using ITK-SNAP. (**A**) WM (blue colour), (**B**) SCC and GCC (red colour), (**C**) deep WM (yellow colour).

**Figure 6 diagnostics-15-01260-f006:**
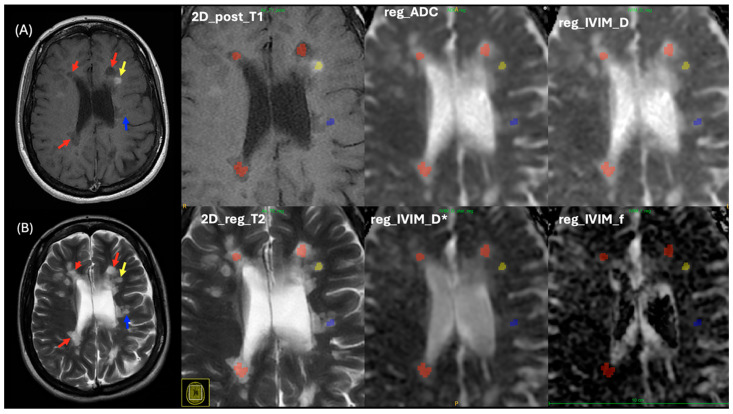
ROIs for three types of MS lesions (enhanced, non-enhanced, and black hole) in a MS patient, delineated on (**A**) T_1_WI as a reference image and (**B**) registered T_2_WI. From the top left to the right, zoom in on the 2D post-contrast T_1_WI, registered ADC map, and registered diffusion coefficient (*D*) map. From down left to right, zoom in registered T_2_WI, registered pseudo-diffusion coefficient (*D**) map, and perfusion fraction (*f*) map. Red arrows and shape indicate black hole MS lesions, Yellow arrows and shapes indicate enhanced MS lesions, blue arrows and shapes indicated non enhanced MS lesions.

**Figure 7 diagnostics-15-01260-f007:**
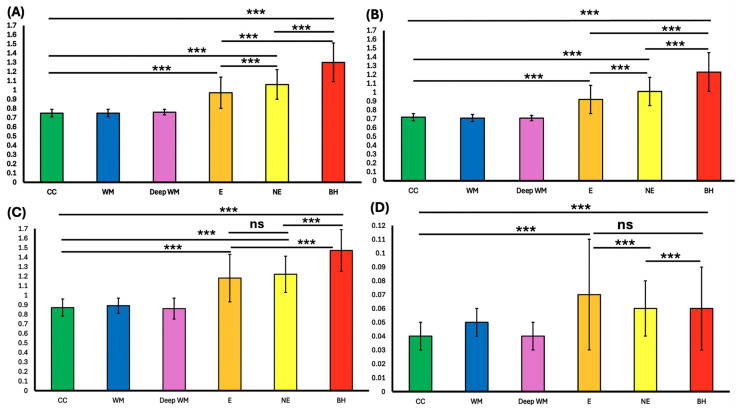
Bar and whisker graphs of (**A**) ADC, (**B**) diffusion coefficient (*D*), (**C**) pseudo-diffusion coefficient (*D**), and (**D**) perfusion fraction (*f*). Data are presented as mean ± standard deviation. *** indicates *p* < 0.001. The units for ADC, *D*, and *D** are 10^−3^ mm^2^/s. ns: “non-significant”.

**Table 1 diagnostics-15-01260-t001:** Demographic and clinical data for controls and RR-MS patients.

	Controls	RR-MS Patients
Sample size and number of examined ROIs in controls and MS patients	29 58 CC, 116 WM, and 116 deep WM A total of 290 collected from different brain areas	224 114 MS_E lesions 1613 MS_NE lesions 779 MS_BH lesions
Mean age ± SD	28 ± 4.7 years	36.3 ± 9.7 years
Gender (male and female)	25 male and 4 female	62 male and 162 female
Disease duration	N/A	8.9 ± 4.7 years
EDSS (mean± SD, range)	N/A	(2.3 ± 1.95, 0–8)
Number of relapses (mean± SD, range)	N/A	(2.5 ± 1.95, 0–13)

**Note:** CC = corpus callosum, WM = white matter BH = black hole, E = enhanced, NE = non-enhanced, EDSS = expanded disability status scale, SD = standard deviation.

**Table 2 diagnostics-15-01260-t002:** Descriptive statistics of IVIM parameters across control regions and MS lesion types.

IVIM Parameters	Groups	N	Mean ± SD
ADC	Corpus callosum	58	0.75 ± 0.04
	WM	116	0.75 ± 0.04
	Deep WM	116	0.76 ± 0.03
	E	114	0.97 ± 0.17
	NE	1613	1.06 ± 0.16
	BH	779	1.30 ± 0.21
*D*	Corpus callosum	58	0.72 ± 0.04
	WM	116	0.71 ± 0.04
	Deep WM	116	0.71 ± 0.03
	E	114	0.92 ± 0.16
	NE	1613	1.01 ± 0.16
	BH	779	1.23 ± 0.22
*D**	Corpus callosum	58	0.87 ± 0.09
	WM	116	0.89 ± 0.08
	Deep WM	116	0.86 ± 0.11
	E	114	1.18 ± 0.25
	NE	1613	1.22 ± 0.19
	BH	779	1.47 ± 0.22
*f*	Corpus callosum	58	0.04 ± 0.01
	WM	116	0.05 ± 0.01
	Deep WM	116	0.04 ± 0.02
	E	114	0.07 ± 0.04
	NE	1613	0.06 ± 0.02
	BH	779	0.06 ± 0.03

**Note:** BH = black hole, E = enhanced, NE = non-enhanced, ADC = apparent diffusion coefficient, *D* = diffusion coefficient, *D** = pseudo-diffusion coefficient, *f* = perfusion fraction, WM = white matter, SD = standard deviation. The units for ADC, *D*, and *D** are 10^−3^ mm^2^/s.

**Table 3 diagnostics-15-01260-t003:** One-way ANOVA of ADC and IVIM parameters comparing control and disease groups.

Outcome Variables		Sum of Squares	df	Mean Square	F-Value	*p*-Value
ADC	Between groups	70.08	5	14.07	482.52	<0.0001
	Within groups	81.04	2790	0.03		
*D*	Between groups	63.08	5	12.62	433.60	<0.0001
	Within groups	81.18	2790	0.03		
*D**	Between groups	80.93	5	16.19	431.40	<0.0001
	Within groups	104.68	2790	0.04		
*f*	Between groups	0.07	5	0.014	25.78	<0.0001
	Within groups	1.55	2790	0.001		

**Note:** ADC = apparent diffusion coefficient, *D* = diffusion coefficient, *D** = pseudo-diffusion coefficient, *f* = perfusion fraction, ANOVA = analysis of variance. Groups are control and MS lesion. The units for ADC, *D*, and *D** are 10^−3^ mm^2^/s.

**Table 4 diagnostics-15-01260-t004:** Multiple comparisons of mean values for ADC and IVIM parameters between control regions and MS lesions.

Healthy Control ROI	MS Lesion	IVIM Parameters	Mean Difference	*p*-Value
Corpus callosum	E	ADC	−0.22	All < 0.0001
	NE		−0.31	
	BH		−0.54	
	E	*D*	−0.20	
	NE		−0.29	
	BH		−0.51	
	E	*D**	−0.31	
	NE		−0.35	
	BH		−0.60	
	E	*f*	−0.02	
	NE		−0.01	
	BH		−0.02	
WM	E	ADC	−0.22	
	NE		−0.31	
	BH		−0.54	
	E	*D*	−0.20	
	NE		−0.30	
	BH		−0.52	
	E	*D**	−0.28	
	NE		−0.32	
	BH		−0.58	
	E	*f*	−0.02	
	NE		−0.01	
	BH		−0.10	
Deep WM	E	ADC	−0.24	
	NE		−0.32	
	BH		−0.56	
	E	*D*	−0.21	
	NE		−0.30	
	BH		−0.52	
	E	*D**	−0.31	
	NE		−0.36	
	BH		−0.61	
	E	*f*	−0.02	
	NE		−0.01	
	BH		−0.02	

**Note:** BH = black hole, E = enhanced, NE = non-enhanced, ADC = apparent diffusion coefficient, *D* = diffusion coefficient, *D** = pseudo-diffusion coefficient, *f* = perfusion fraction, WM = white matter. The units for ADC, *D*, and *D** are 10^−3^ mm^2^/s.

**Table 5 diagnostics-15-01260-t005:** Multiple comparisons of mean values for diffusion and IVIM parameters between different MS lesions.

IVIM Parameters	MS Lesions		Mean Difference	*p*-Value
ADC	E	NE	−0.09	All < 0.0001
	E	BH	−0.32	
	NE	BH	−0.23	
*D*	E	NE	−0.09	All < 0.0001
	E	BH	−0.22	
	NE	BH	−0.22	
*D**	E	NE	−0.04	0.264
	E	BH	−0.30	Both < 0.0001
	NE	BH	−0.25	
*f*	E	NE	0.01	<0.0001
	E	BH	0.01	0.205
	NE	BH	−0.005	<0.0001

**Note:** BH = black hole, E = enhanced, NE = non-enhanced, ADC = apparent diffusion coefficient, *D* = diffusion coefficient, *D** = pseudo-diffusion coefficient, f = perfusion fraction, WM = white matter. The units for ADC, *D*, and *D** are 10^−3^ mm^2^/s.

**Table 6 diagnostics-15-01260-t006:** Cutoff values of outcome variables (ADC, D, D*, f) using receiver operating characteristic curve analysis.

Type of MS vs. Controls	Cutoff Value for Detecting Lesions	Area Under the Curve	*p*-Value	Sensitivity (95% CI)	Specificity (95% CI)
ADC					
BH vs. Controls	>0.84	0.996	All < 0.001	98.0 (96.6–98.7)	99.7 (97.9–100)
E vs. Controls	>0.83	0.967		89.5 (82.3–94.4)	98.6 (96.5–99.6)
NE vs. Controls	>0.83	0.989		94.5 (93.3–95.5)	98.3 (96.0–99.4)
*D*					
BH vs. Controls	>0.81	0.996	All < 0.001	97.7 (96.4–98.6)	99.6 (97.5–99.9)
E vs. Controls	>0.78	0.948		89.5 (82.3–94.4)	96.6 (93.8–98.3)
NE vs. Controls	>0.79	0.982		93.6 (92.2–94.7)	98.3 (96.0–99.4)
*D**					
BH vs. Controls	>1.08	0.99	All < 0.001	95.3 (93.5–96.6)	99.0 (97.0–99.8)
E vs. Controls	>0.98	0.917		86.0 (78.2–91.8)	89.7 (85.6–92.9)
NE vs. Controls	>1.00	0.957		88.5 (86.8–90.0)	92.8 (89.1–95.5)
*f*					
BH vs. Controls	>0.05	0.715	All < 0.001	56.7 (53.2–60.3)	80.0 (74.2–83.8)
E vs. Controls	>0.05	0.728		61.4 (51.8–70.4)	80.7 (75.7–85.1)
NE vs. Controls	>0.05	0.67		49.8 (47.3–52.2)	80.0 (74.9–84.4)

**Note:** BH = black hole, E = enhanced, NE = non-enhanced, ADC = apparent diffusion coefficient, *D* = diffusion coefficient, *D** = pseudo-diffusion coefficient, *f* = perfusion fraction, WM = white matter. The units for ADC, *D*, and *D** are 10^−3^ mm^2^/s.

## Data Availability

The data presented in this study are available on request from the corresponding author, as they are currently being used in another clinical experiment.

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
