# Peer review of "Investigating the Role of Intravoxel Incoherent Motion Diffusion-Weighted Imaging in Evaluating Multiple Sclerosis Lesions"

_diagnostics, 2025, doi:10.3390/diagnostics15101260_

Round 1
Reviewer 1 Report (Previous Reviewer 1)
Comments and Suggestions for Authors
I appreciate the authors' efforts to improve this article, however Results' sections should be improved.
3.3 Multiple Comparisons of MRI Parameters
3.3.1. Control Regions vs. MS Lesions
Table 4, 5 show a p-values, but it is not clear if authors refer to Tukey’s tests or ANOVA.
They should clarify these issues.
P-values were not corrected for multiple comparisons due to the number of ROIs, however they are so low to remain statistically significant.
Author Response
|

Reviewer 2 Report (New Reviewer)
Comments and Suggestions for Authors
Brief Summary
The manuscript aims to investigate the potential of IVIM parameters in diagnosing brain lesions of MS and discriminating between enhanced, non-enhanced, and black-hole lesions, potentially addressing the need for gadolinium injection. The authors demonstrate the relatively high sensitivity and specificity of these features by investigating 237 MS cases and 29 controls.
General Comments
The use of the English language is appropriate, and the methods used in the study are well described. The study has the potential to provide a breeding ground for the validation of IVIM DWI by larger studies as a substitute for contrast-enhanced MRI, avoiding the side effects of gadolinium. However, there are inherent limitations in terms of the small number of controls and the lack of the recruitment of other patients with similar brain lesions yet distinct disorders (notably neuromyelitis optica spectrum disorders [NMOSD]). By acknowledging these limitations, I believe that the manuscript would be eligible for publication after minor revisions.
Major Issues
- 4, Paragraph 2: It is mentioned that 13 of 237 cases were excluded due to the suboptimal image quality, yet I still see all 237 cases in the subsequent tables and analyses. This must be clarified.
- The authors should hypothesize why IVIM DWI has failed to discriminate all MS lesions from normal brain tissue.
- The limitations of the study (including the lack of comparison with other disorders with a similar MRI picture) need to be clearly stated.
Minor Issues
- P. 3, l. 132: Exact dates (years as well as months) should be declared.
Author Response
|
Comments 1: 4, Paragraph 2: It is mentioned that 13 of 237 cases were excluded due to the suboptimal image quality, yet I still see all 237 cases in the subsequent tables and analyses. This must be clarified. |
|
Response 1: I appreciate your important comment. It was a typing mistake. The correct patient number is 224, and we have made the following updates for the patient number and for both male and female patients in the study:
Abstract (Methods): (2,506 lesions from 224 patients with MS; excluding 13 patients due to MRI artifacts or not meeting the diagnostic criteria for RR-MS) These changes can be found on page number 1, paragraph, and lines 32 and 33.
In table 1, Sample size has been corrected to 224. The correct number of male and female samples are 62 and 162, respectively.
These changes can be found on page number 4, table 1, and lines 159 and 160.
Please note that all statistical results remain unchanged, as our analyses are based on the number of ROIs as the sample size; therefore, no further changes are required for other tables or graphs.
|
|
Comments 2: The authors should hypothesize why IVIM DWI has failed to discriminate all MS lesions from normal brain tissue. |
|
Response 2: Thank you for the insightful suggestion. Figure 7 and Table 4 show statistically significant differences (p < 0.0001) in IVIM-derived parameters (ADC, D, D*, and f) between control ROIs and each MS lesion category at the group level. However, to address potential limitations at the individual-lesion level, the following paragraph has been added to the end of the Discussion:
Although our cohort analysis demonstrates robust discrimination of MS types from the control group (Table 4), individual lesion-level accuracy may be constrained by several factors. First, MS plaques progress pathologically from acute inflammatory to chronic demyelinated or remyelinated stages, resulting in diffusion-perfusion patterns that may overlap with those of control tissue [51]. Second, partial volume effects in small or irregularly shaped plaques can distort parameter contrasts [52]. Finally, the biexponential IVIM model is sensitive to noise at high b-values and is limited by a low-b sampling scheme, which can impair the precision of D and f estimates in lesions with subtle microvascular changes [21].
51. Lucchinetti CF, Brück W, Parisi JE et al., Heterogeneity of multiple sclerosis lesions: implications for the pathogenesis of demyelination. Ann.Neurol. 2000; 47:707–717.
52. Roine T, Jeurissen B, Perrone D et al., Isotropic non-white matter partial volume effects in constrained spherical deconvolution. Front. Neuroinform. 2014; 8.
These changes can be found on page number 17., paragraph 2 and lines number 516 to 524. Comments 3: The limitations of the study (including the lack of comparison with other disorders with a similar MRI picture) need to be clearly stated. Response 3: Thank you for your valuable comments. We have included a paragraph at the end of the discussion section. In our study, MS patients had been carefully selected based on the McDonald criteria [11, 34], and without other detectable neurological disorders. There may be, however, some overlaps of IVIM signatures in MS lesions compared to brain tumor lesions or other lesions mimicking MS [53]. Future studies could investigate the specificity of IVIM for differentiating MS lesions from other types of brain lesions [54, 55].
53. Preziosa P, Sangalli F, Esposito F et al., Clinical deterioration due to co-occurrence of multiple sclerosis and glioblastoma: report of two cases. Neurol. Sci. 2017; 38:361-364.
54. He T, Zhao W, Mao Y et al., MS or not MS: T2-weighted imaging (T2WI)-based radiomic findings distinguish MS from its mimics. Mult. Scler. Relat. Disord .2022; 61: 10376.
55. Elkossi SM, Sayed SA, Shehata GA et al., Diffusion tensor imaging and voxel-based morphometry in differentiating multiple sclerosis and its mimics. EJRNM. 2023; 54: 216.
These changes can be found on page number17 , paragraph number 3 and lines number 526 to 531. Comments 4: P. 3, l. 132: Exact dates (years as well as months) should be declared. Response 4: I appreciate your careful review of the manuscript. The exact dates were in included in the manuscript. “This prospective study was conducted from January 2019 to December 2020”.
These changes can be found on page number 3 , paragraph number 5 and lines number 131 and 132.
|

This manuscript is a resubmission of an earlier submission. The following is a list of the peer review reports and author responses from that submission.
Round 1
Reviewer 1 Report
Comments and Suggestions for Authors
I appreciated the great effort of the authors for this paper but there are some critical points that the need to be corrected and improved.
The main points are as follows
2.4. MRI Processing
The authors should explain why they did not perform sample size for each variables and ROIs examined.
In my opinion each groups ROIs should be outlined using a high resolution 3D imaging sequence.
Moreover authors should clarify how lesions ROIs were registered to parametric maps for MS lesions.
2.5. Statistical analysis
No multiple comparison corrections were performed taking into account the number of ROIs and the parameters evaluated.
3.1 Descriptive Statistics
It is really difficult to understand ROIs’ distributions values of each parameter in table 1 without figures.
Author Response
|
Comments 1: The main points are as follows: 2.4. MRI Processing. The authors should explain why they did not perform sample size for each variable and ROIs examined.
|
|
Response 1: As our study is the first for IVIM to characterize MS lesions, there was no prior information to use towards the calculation of sample size. To answer the reviewer’s question, we performed a post-hoc power calculation to test the minimum sample size using reference ADC values [1], as follows: The ADC of various brain WM structures in MS patients were shown to be elevated (0.79 ± 0.02 × 10-3 mm/s) compared to the control group (0.75 ± 0.02 × 10-3 mm/s) [1]. Assuming that IVIM would have similar power statistics with the ADC, we calculated the sample size using G*Power 3.1 [2]. Using input parameters: tails = 2, effect size d = 1.57 (calculated from the reference ADC values [1]), a error probability = 0.05, power (1-b error probability) = 0.8, a minimum sample size of 8 was estimated to be required in each group. If we increase the power level to 0.95, a sample size of 12 was required in each group.
Our study used data from 237 MS and 29 healthy control subjects, which are higher than the calculated minimum sample size. For comparison of healthy vs MS groups, we used a much larger number of ROIs to enable more accurate analysis of MS lesion types in different areas of the brain. Our results showed highly significant statistical difference of the parameters between the groups, indicating that a sufficient power and sample number had been used in this study. To clarify the use of ROIs for statistics, the following changes were made in the 2. Materials and Methods, 2.1. Sample Criteria: The number of sample size in our experiment was assumed to be the number of regions of interest (ROIs). This is a reasonable assumption because it permits comparing between similarly located ROIs in healthy subjects and MS, as the lesions were found to aggregate in various brain structures [28]. Such an approach was found useful to improve the assessment accuracy and better understanding of microstructural changes in both healthy tissues and all MS lesions, such as axonal loss [29, 30].
Our experiment included 29 controls with the IVIM parameters measured using a total of 290 ROIs in various brain structures, including the corpus callosum (CC), white matter (WM), and deep WM. These ROIs were chosen to compare to types of MS lesions in specific brain areas, which are highly myelinated and common sites for MS lesions [31, 32]. The ROI selection in the CC allowed evaluation of brain connectivity between interhemispheric structure [33] and provided a reference for evaluating either demyelination or axonal injury of the nervous tissues [29, 30].
The composition of ROIs in controls are as follows: 58 ROIs in the CC, comprised of two ROIs at the genu and splenium. 116 ROIs in the WM, comprised of two ROIs in the right and two ROIs in the left frontal WM areas. 116 ROIs in the deep WM, comprised of four ROIs located in the right and the left thalamus, and in the right and the left caudate nuclei.
These updated paragraphs can be found–page number 3 and 4, 2nd paragraph, and 3rd paragraph in the 2. Materials and Methods, 2.1. Sample Criteria, and line 132-148]
|
|
Comments 2: In my opinion, each group's ROIs should be outlined using a high-resolution 3D imaging sequence.
|
|
Response 2: We agree that 3D high-resolution imaging would enable detection of MS lesions more accurately, as previously 3D pre- and post-contrast T1 WI and 3D FLAIR had been found to be useful in detecting more MS lesions, particularly in the cortical and subcortical areas [3-5]. Unfortunately, we are unable to incorporate outlining ROIs using high-resolution 3D imaging due to data limitations. For the control data, we did not perform 3D acquisitions due to the time constraints imposed by the hospital on our MRI scanners for scanning healthy volunteers. Therefore, we only performed the analyses using registrations of 2D data so that the processing was identical for comparison between the patients and the controls. This limitation has been added in the discussion section of the manuscript, as follows: Third, 3D acquisitions were not used to delineate ROI MS lesions in our experiment because no 3D images were obtained for control subjects due to time constraints when using clinical scanners. This is one of the study's limitations, as 3D pre- and post-contrast T1 WI and 3D FLAIR would help detect more MS lesions, particularly in the cortical and subcortical areas [48-50]. This could be used in future studies to classify MS lesions and quantify IVIM values in all brain regions. Last, the smaller control group sample size was primarily a practical limitation due to the challenges in recruiting healthy volunteers for advanced MRI protocols. Nevertheless, the preliminary findings from this study provide a valuable foundation for future work, where we aim to increase the control sample size to validate and expand upon these results.
These updated paragraphs can be found– page number 16 and 17, last paragraph in the discussion section, and line 503 to 512] Comments 3: Moreover, authors should clarify how lesions ROIs were registered to parametric maps for MS lesions. Response 3: The following paragraphs detail how the IVIM parametric maps were registered for both control subjects and MS patients. DWI datasets were acquired using 2D EPI at 0.8 × 0.8 × 5 mm resolution and 2D T1- and T2-weighted images were acquired at 0.4 × 0.4 × 4.5 mm resolution. DWI had a lower image resolution than the T1WI and T2WI to minimize EPI distortion artefacts. DWI motion correction was applied by the scanner during the scan. For each patient, the ADC and IVIM maps derived from the DWI and b0 images were already co-registered.
Image registration was performed on all patient data to ensure all MR images were correctly aligned. This is important because some patients were repositioned after contrast media administration (Figure 3), and image registration would increase the accuracy of defining MS lesions and facilitated comparison between the T1WI and T2WI, as well as the IVIM parametric maps.
Post-contrast 2D T1WI was used as a reference to register the image dataset for each patient. Firstly, the T2WI was registered to T1WI using a rigid body followed by an affine registration. Secondly, the T2 WI was then registered to DWI’s b0 image as they had similar contrast. This transformation was then applied to the T1WI, so that all images were all in the same space. A correlation ratio cost function was applied for the multi-contrast MRI registration [36-38], and a tri-linear image interpolation was used to smooth the registered images for analysis. This step was critical in identifying three types of MS lesions: enhanced, non-enhanced, and black hole.
Figure 3. Pre- and post-linear registration of the IVIM parametric maps of MS patient (A) T2 WI, (B) ADC map, (C) Diffusion Coefficient (D) Map (D) Pseudo-Diffusion Coefficient (D*) Map, and (E) Perfusion Fraction (f) Map to the T1WI as a reference image (F).
A similar registration approach was utilized for both control subjects and MS patients. However, for control subjects, we used b0 as a reference image because 2D T1WI and T2WI were not acquired for healthy participants. Figure 4 demonstrates the registration of b0 to all IVIM parametric maps. Figure 4. Pre- and post-linear registration of the IVIM parametric maps of control subject (A) ADC map, (B) Diffusion Coefficient (D) Map (C) Pseudo-Diffusion Coefficient (D*) Map, and (D) Perfusion Fraction (f) Map to the b0 as a reference image (E).
Linear registration FLIRT FSL helped to limit rotation, scaling, shearing, and translation, which might occur due to patient movement or image distortion. This registration method was appropriate because all images were registered within the same subject. This in turn improved the experiment's workflow and ROI selection in the same 2D structural images and 2D IVIM parametric maps for both control and MS patients. This consequently allowed us to quantify ADC, D, D*, and f values using the IVIM parametric maps in both control and MS patients by reading out all ROIs using ITK-SNAP [39], Figure 5 and 6.
These updated paragraphs can be found– page number 6 to 8., paragraph 3, 4 and 5 in 2.4 MRI processing, and line 218 to 255]
Figure 5. Regions of interest (ROIs) for control participants, delineated on b0 images using ITK-SNAP.
|
|
Figure 6. ROIs for three types of MS lesions (enhanced, non-enhanced, and black hole) in a MS patient, delineated on (A) T1WI as a reference image and (B) registered T2WI. From the top left to the right, zoom in on the 2D post-contrast T1WI, registered ADC map, and registered Diffusion Coefficient (D) Map. From down left to right, zoom in registered T2WI, registered Pseudo-Diffusion Coefficient (D*) Map, and Perfusion Fraction (f) Map
These updated paragraphs and figures can be found– page number 8 and 9, figure 5 and 6, and line 268 to 283]
Comments 4: 2.5. Statistical analysis No multiple comparison corrections were performed taking into account the number of ROIs and the parameters evaluated. Response 4: No need for corrections as the post hoc comparison test had taken into account the corrections. Comments 5: 3.1 Descriptive Statistics It is really difficult to understand ROIs’ distributions values of each parameter in table 1 without figures. Response 5: We agree that it is better to present Table 1 as a graph chart, which demonstrates how the ADC, D, D*, and f are distributed and the relationship between ROIs of the control subjects and ROIs of the MS lesions.
Figure 7. Bar and whisker graphs of (A) ADC, (B) Diffusion Coefficient (D) (C) Pseudo-Diffusion Coefficient (D*) and (D) Perfusion Fraction (f). Data were presented as mean ± standard deviation. *** indicates p < 0.001. The units for ADC, D and D* are 10-3 mm2/s.
This updated figure can be found– page number 11, and line 329]
Reference for the responses
1. Faul F, Erdfedler E Lang A-G et al. G*Power 3: a flexible statistical power analysis program for the social, behavioral, and biomedical sciences. Behav Res Methods, 2007; 39:175-91. 2. Zacharzewska-Gondek A, pokryszko-Dragan A, Maciej Gondek T et al. Apparent diffusion coefficient measurements in normal appearing white matter may support the differential diagnosis between multiple sclerosis lesions and other white matter hyperintensities. J Neurol Sci. 2019; 397:24-30. 3. Kummari S, Burra KG, Reddy VR, Das S et al. Determination of Efficiency of 3D Fluid-Attenuated Inversion Recovery (FLAIR) in the Imaging of Multiple Sclerosis in Comparison With 2D FLAIR at 3-Tesla MRI. Cureus 2023; 15:e48136. 4. Filippi M, Rocca MA, Calabrese M et al. Intracortical lesions: relevance for new MRI diagnostic criteria for multiple sclerosis. Neurology 2010; 75:1988-94. 5. Geurts JJ, Pouwels PJ, Uitdehaag BM et al., Intracortical lesions in multiple sclerosis: improved detection with 3D double inversion-recovery MR imaging. Radiology 2005; 236:254-60.
5. Additional clarifications |
|
1. Additional references were included in the reference section of the manuscript.
2. All the references citation in the manuscript were renumbered according to the updated reference list
3. All the changes in the manuscript are marked in red color.
4. In the page number 13, line 379 and 380, there is missing sentence. It is included in the manuscript as the following: …
“The following values were determined to distinguish between BH vs controls, E vs controls and non-enhanced vs controls, respectively” For ADC, the cut-off values were > 0.84, > 0.83, > 0.83 with sensitivity 98.0%, 89.5%, 94.5% and specificity 99.7%, 98.6%, 98.3%.
|

Reviewer 2 Report
Comments and Suggestions for Authors
This research article investigates the use of intravoxel incoherent motion (IVIM) diffusion and perfusion MRI to analyze multiple sclerosis (MS) lesions. Researchers compared IVIM metrics (ADC, D, D*, and f) in various MS lesion types (enhanced, non-enhanced, black hole) against healthy brain tissue in a large cohort of patients and controls. The study found significant differences in IVIM parameters among the groups, suggesting these metrics could serve as valuable diagnostic biomarkers for MS. Statistical analyses, including ANOVA and ROC curve analysis, supported the high sensitivity and specificity of IVIM in differentiating MS lesions from healthy tissue and in distinguishing between lesion subtypes. The findings suggest that IVIM MRI offers a potentially safer and more informative alternative to gadolinium-based contrast agents in MS imaging.
The study is overall sound, with a clear objective to assess diffusion and perfusion MRI metrics across different brain regions in healthy individuals and various types of MS lesions.
However, I have some major concerns that should be addressed.
1. Using only 29 healthy controls compared to 237 patients with MS can reduce the statistical power of the study, as a smaller control group may limit the ability to detect true differences between groups and increase the risk of Type II errors. The imbalance in sample sizes can also introduce potential bias and affect the generalizability of the findings, as the control group may not adequately represent the broader population. Additionally, wider confidence intervals for the control group due to the smaller sample size can lead to less precise estimates.
2. I noticed the absence of demographic tables detailing participant characteristics. Including such tables is essential for papers of this nature to present all subject-related data clearly and concisely.
3. The "Conclusions" section is not correct, does not reflect your findings, and kindly needs to be revised.
Minor:
1. The first use of each abbreviation should be defined such as "**MS** is a chronic and heterogeneous ..."
2. Kindly specify the number of the channel coils: "multichannel phased-array head coil..."
Author Response
|
Comments 1: Using only 29 healthy controls compared to 237 patients with MS can reduce the statistical power of the study, as a smaller control group may limit the ability to detect true differences between groups and increase the risk of Type II errors. The imbalance in sample sizes can also introduce potential bias and affect the generalizability of the findings, as the control group may not adequately represent the broader population. Additionally, wider confidence intervals for the control group due to the smaller sample size can lead to less precise estimates. |
||||||||||||||||||||||
|
Response 1: We appreciate the reviewer's valuable feedback on the control group's sample size. We would like to emphasize that this study is a preliminary investigation into the feasibility and potential of DWI and IVIM parameters for distinguishing between different types of MS lesions and normal tissue. As such, the findings should be interpreted as a critical step in hypothesis generation, guiding future studies with larger and more balanced sample sizes. As our study is the first for IVIM to characterize MS lesions, there was no prior information to use towards the calculation of sample size. To answer the reviewer’s question, we performed a post-hoc power calculation to test the minimum sample size using reference ADC values [1], as follows: The ADC of various brain WM structures in MS patients were shown to be elevated (0.79 ± 0.02 × 10-3 mm/s) compared to the control group (0.75 ± 0.02 × 10-3 mm/s) [1]. Assuming that IVIM would have similar power statistics with the ADC, we calculated the sample size using G*Power 3.1 [2] . Using input parameters: tails = 2, effect size d = 1.57 (calculated from the reference ADC values [1]), a error probability = 0.05, power (1-b error probability) = 0.8, a minimum sample size of 8 was estimated to be required in each group. If we increase the power level to 0.95, a sample size of 12 was required in each group. Our study used data from 237 MS and 29 healthy control subjects, which are higher than the calculated minimum sample size. For comparison of healthy vs MS groups, we used a much larger number of ROIs to enable more accurate analysis of MS lesion types in different areas of the brain. Our results showed highly significant statistical difference of the parameters between the groups, indicating that a sufficient power and sample number had been used in this study. Furthermore, we used a robust methodology to make the most of the available data from the control group. We specifically used three ROIs types (each consisted of 2-4 individual ROIs) per control participant to increase statistical power. We made the following changes in our manuscript in the 2. Materials and Methods, 2.1. Sample Criteria:
The number of sample size in our experiment was assumed to be the number of regions of interest (ROIs). This is a reasonable assumption because it permits comparing between similarly located ROIs in healthy subjects and MS, as the lesions were found to aggregate in various brain structures [28]. Such an approach was found useful to improve the assessment accuracy and better understanding of microstructural changes in both healthy tissues and all MS lesions, such as axonal loss [29, 30]. Our experiment included 29 controls with the IVIM parameters measured using a total of 290 ROIs in various brain structures, including the corpus callosum (CC), white matter (WM), and deep WM. These ROIs were chosen to compare to types of MS lesions in specific brain areas, which are highly myelinated and common sites for MS lesions [31, 32]. The ROI selection in the CC allowed evaluation of brain connectivity between interhemispheric structure [33] and provided a reference for evaluating either demyelination or axonal injury of the nervous tissues [29, 30]. The composition of ROIs in controls are as follows: 58 ROIs in the CC, comprised of two ROIs at the genu and splenium. 116 ROIs in the WM, comprised of two ROIs in the right and two ROIs in the left frontal WM areas. 116 ROIs in the deep WM, comprised of four ROIs located in the right and the left thalamus, and in the right and the left caudate nuclei.
These updated paragraphs can be found–page number 3 and 4, 2nd paragraph, and 3rd paragraph in the 2. Materials and Methods, 2.1. Sample Criteria, and line 132-148]
While the sample sizes of the control and patient groups are indeed imbalanced, the large patient cohort provides reliable estimates for the MS population, while the smaller control group, though limited, is sufficient for identifying key differences between groups as shown in Tables 1-4. Finally, the smaller control group sample size was primarily a practical limitation due to the challenges in recruiting healthy volunteers for advanced MRI protocols, and time limitation imposed by the Hospital in spending time to scan healthy controls. Nevertheless, the preliminary findings from this study provide a valuable foundation for future work, where we aim to increase the control sample size to validate and expand upon these results. Your comments and suggestions helped us to improve the manuscript. The limitation paragraph in the discussion has been updated in response to your comments and suggestions as follows: Third, 3D acquisitions were not used to delineate ROI MS lesions in our experiment because no 3D images were obtained for control subjects due to time constraints when using clinical scanners. This is one of the study's limitations, as 3D pre- and post-contrast T1 WI and 3D FLAIR would help detect more MS lesions, particularly in the cortical and subcortical areas [48-50]. This could be used in future studies to classify MS lesions and quantify IVIM values in all brain regions. Last, the smaller control group sample size was primarily a practical limitation due to the challenges in recruiting healthy volunteers for advanced MRI protocols. Nevertheless, the preliminary findings from this study provide a valuable foundation for future work, where we aim to increase the control sample size to validate and expand upon these results. These updated paragraphs can be found– page number 16 and 17 , last paragraph in the discussion section, and line 503 to 512]
|
||||||||||||||||||||||
|
Comments 2: I noticed the absence of demographic tables detailing participant characteristics. Including such tables is essential for papers of this nature to present all subject-related data clearly and concisely. |
||||||||||||||||||||||
|
Response 2: We agree that presenting this demographic table will enhance the presentation of our paper. This has been added as follows: Table 1 Controls and RR-MS patients Demographic and clinical data
Note: CC= corpus callosum, WM =white matter BH = Black Hole, E= Enhanced, NE = Non-Enhanced, EDSS=Expanded disability status scale SD = standard deviation.
These updated paragraphs can be found– page number 4, paragraph 5 in the materials and methods section, and line 158-160]
Comments 3: The "Conclusions" section is not correct, does not reflect your findings, and kindly needs to be revised. Response 3: We revised the conclusion to better reflect the findings of the study. Our study showed that the ADC and IVIM D parameters were sensitive and specific markers to distinguish between normal white matter brain tissues and different types of MS lesions. The role of perfusion IVIM D* and IVIM f, however, appeared to be more limited. Further studies are needed to determine if ADC and IVIM can reliably be used, instead of contrast agents, to distinguish normal brain tissue from different MS lesion types.
These updated paragraphs can be found– page number 17, 1st paragraph in the conclusion section, and line 511 to 516] Comments 4: 1. The first use of each abbreviation should be defined such as "**MS** is a chronic and heterogeneous ..." Response 4: We reviewed the first use of each abbreviation and corrected it accordingly in multiple places. Multiple sclerosis (MS) is a chronic and heterogeneous disease characterised by demyelination and axonal loss and damage. These updated paragraphs can be found– page number 1 1st paragraph in the abstract section, and line 20] Magnetic Resonance Imaging (MRI) has been employed to distinguish these changes in various types of MS lesions. These updated paragraphs can be found– page number 1 1st paragraph in the abstract section, and line 21-22]
The imaging sequences included three-dimensional (3D) T1, 3D fluid-attenuated inversion recovery, two-dimensional (2D) T1, T2-weighted imaging, and 2D diffusion-weighted imaging (DWI) sequences.
These updated paragraphs can be found– page number 1 in the abstract section, and line 27-30]
Most patients with MS are diagnosed with relapsing-remitting multiple sclerosis (RR-MS), which primarily affects young adults, with an average age of symptom onset of 30 years and a higher incidence in women.
These updated paragraphs can be found– page number 2, 2nd paragraph in the introduction section, and line 57 to 58]
Comments 5: 2. Kindly specify the number of the channel coils: "multichannel phased-array head coil..." Response 5: In the method section, this has been fixed as "multichannel phased-array head coil equipped with 8 channels.”
These updated paragraphs can be found– page number 5, in the first paragraph in section 2.3. MRI Imaging Protocols, and line 181-182]
|
